# The telomere length landscape of prostate cancer

Julie Livingstone [1,2,3,4], Yu-Jia Shiah[5], Takafumi N. Yamaguchi[1,2,3,4], Lawrence E. Heisler[5], Vincent Huang [5], Robert Lesurf [5], Tsumugi Gebo[1,2,3,4], Benjamin Carlin[1,2,3,4], Stefan Eng[1,2,3,4], Erik Drysdale[5], Jeffrey Green[5], Theodorus van der Kwast [6,7], Robert G. Bristow [6,8,9], Michael Fraser [6] & Paul C. Boutros [1,2,3,4,10] ✉

Replicative immortality is a hallmark of cancer, and can be achieved through telomere lengthening and maintenance. Although the role of telomere length in cancer has been well studied, its association to genomic features is less well known. Here, we report the telomere lengths of 392 localized prostate cancer tumours and characterize their relationship to genomic, transcriptomic and proteomic features. Shorter tumour telomere lengths are associated with elevated genomic instability, including single-nucleotide variants, indels and structural variants. Genes involved in cell proliferation and signaling are correlated with tumour telomere length at all levels of the central dogma. Telomere length is also associated with multiple clinical features of a tumour. Longer telomere lengths in non-tumour samples are associated with a lower rate of biochemical relapse. In summary, we describe the multi-level integration of telomere length, genomics, transcriptomics and proteomics in localized prostate cancer.

[1] Department of Human Genetics, University of California, Los Angeles, CA 90095, USA. [2] Department of Urology, University of California, Los Angeles, CA 90024, USA. [3] Jonsson Comprehensive Cancer Centre, University of California, Los Angeles, CA 90024, USA. [4] Institute for Precision Health, University of California, Los Angeles, CA 90024, USA. [5] Ontario Institute for Cancer Research, Toronto, ON M5G 0A3, Canada. [6] Princess Margaret Cancer Centre, University Health Network, Toronto, ON M5G 2M9, Canada. [7] Department of Pathology, Laboratory Medicine Program, University Health Network, Toronto, ON M5G 2C4, Canada. [8] Department of Medical Biophysics, University of Toronto, Toronto, ON M5G 1L7, Canada. [9] Manchester Cancer Research Centre, Manchester, UK. [10] Department of Pharmacology and Toxicology, University of Toronto, Toronto, ON M5S 1A8, Canada. ✉email: pboutros@mednet.ucla.edu

 

Telomeres, which make up the ends of chromosomes, consist of a repeat TTAGGG sequence[1] along with bound proteins known as shelterin[2]. Telomeres protect chromosomal ends from degradation by the DNA double-strand break (DSB) response pathway. Due to the linearity of chromosomes and chromosomal replication, telomeres are shortened by approximately 50 bp during mitosis[3]. When telomeres become substantially shortened, cell cycle progression halts and cells enter replicative senescence; further replication leads to cellular crisis and eventually cell death[4]. Telomere maintenance and lengthening is essential for cancer cell proliferation and enables replicative immortality: a fundamental hallmark of cancer[5]. Telomere regulation occurs through two known mechanisms: activation of telomerase or alternative lengthening of telomeres (ALT) which relies on homology-directed DNA replication[6].

Despite the pan-cancer studies analysing telomere length from various tumour types[7,8], the role of telomere maintenance in individual tumour types is poorly understood. Moreover, the relationship between telomere length and biologically relevant genomic indices, such as percentage of the genome altered (PGA;[9,10]), and other measures of mutational density has not been assessed, nor has the association between telomere length and clinical outcome in prostate cancer.

We and others have described the genomic, transcriptomic and proteomic landscape of localized, non-indolent prostate cancer:[11–18] the most frequently diagnosed non-skin malignancy in North American men (~250,000 new cases per year). Localized prostate cancer is a C-class tumour[19], characterized by a paucity of driver single-nucleotide variants (SNVs) and a relatively large number of structural variants (SVs), including copy number aberrations (CNAs) and genomic rearrangements (GRs). Several of these aberrations, including mutations in *ATM* and amplifications of *MYC* – which drive DSB repair and cell proliferation, respectively—are associated with significantly reduced time to biochemical and metastatic relapse after local therapy[20]. Intriguingly, both of these mutations have also been associated with telomere maintenance[21,22] and telomere shortening–relative to adjacent epithelium[23]. Similarly, an interaction between hypoxia, dysregulated *PTEN*, *TERT* abundance and telomere shortening were recently illustrated[15]. Despite this, no well-powered study exists evaluating the association between telomere length, somatic features and clinical outcome in prostate cancer.

To fill this gap, we quantify the telomere length and somatic mutational landscapes of 392 localized prostate tumours. We explore associations between telomere length and the tumour methylome, transcriptome and proteome. Using rich clinical annotation, we further assess the relationship between telomere length and outcome. Taken together, these data establish the role and regulation of telomere length in localized prostate cancer, and establish clear links between telomere maintenance and drivers of prostate cancer development and clinical aggression.

## Results

**Association of telomere length with somatic nuclear driver events**. To investigate the impact of telomere length (TL) on the clinico-genomics of prostate tumours, we exploited whole genome sequencing (WGS) of 392 published tumour–reference pairs[11–14,24]. We estimated both tumour and non-tumour (blood or adjacent histologically normal tissue) TLs for each sample using TelSeq v0.0.1[25] and TelomereHunter v1.0.4[26]. After quality control, 381 samples were retained for further analysis (see Methods). All tumours were treatment-naive, and detailed clinical information was collected. The cohort consisted of 11.2% ISUP Grade Group (GG) 1, 52.8% GG2, 24.6 % GG3, 7.8% GG4 and 3.4% GG5. For the majority of samples, the tumour was confined

to the prostate (6.5% T1, 53.0% T2, 40.0% T3, 0.5% T4). The mean tumour coverage was 73.1x ± 20.6x (median ± standard deviation); the mean non-tumour coverage was 44.1x ± 13.4x. Median clinical follow-up time was 7.46 years. TLs for each sample, along with clinical and genomic summary data are in Supplementary Data 1. Non-tumour TLs varied dramatically across individuals, ranging from 2.10 kbp to 15.0 kbp, with a median of 4.52 ± 1.35 kbp. Adjacent normal TLs ($n = 40$) were longer than those in blood tissue ($n = 341$; $P = 2.80 \times 10^{-10}$; two-sided Mann–Whitney $U$ test) and tumour tissue ($P = 3.04 \times 10^{-21}$; two-sided Mann–Whitney $U$ test; Supplementary Fig. 1a). By contrast, tumour TLs varied less but were significantly shorter, ranging from 1.03 to 6.45 kbp with a median of 3.36 ± 0.87 kbp. Non-tumour TLs were not associated with sequencing coverage (Supplementary Fig. 1b). Tumour TLs were independent of tumour purity but there was a weak negative correlation between coverage and TL driven by some samples sequenced with over 100x coverage (Fig. 1d; Supplementary Figs. 1b-c). Tumour and non-tumour TL estimates from TelSeq and TelomereHunter were highly correlated (Supplementary Fig. 1d) so we decided to use TelSeq estimates throughout. There was no difference in TL ratio between localized and metastatic samples ($n = 101$; $P = 0.95$; two-sided Mann–Whitney $U$ test; Supplementary Fig. 1e). To account for batch effects and the differences in blood and normal adjacent tissue, a linear model was fit and TLs were adjusted (Supplementary Figs. 1f, g). TL ratios (tumour TL/non-tumour TL) were calculated to further reduce any confounding effects of the sequencing method. Tumour and non-tumour TLs were positively correlated with one another ($\rho = 0.37$, $P = 7.30 \times 10^{-14}$, Fig. 1a). As expected, TL ratio was positively correlated with tumour TL ($\rho = 0.63$, $P < 2.2 \times 10^{-16}$; Fig. 1b) but negatively correlated with non-tumour TLs ($\rho = -0.40$, $P < 2.2 \times 10^{-16}$; Fig. 1c).

To assess whether tumour TL was related to any specific genomic property of a tumour, we evaluated a set of driver mutations previously identified in prostate cancer[14]. The relationship of each of these features with tumour TL is shown in Fig. 1d. While tumour TL was not associated with any known prostate cancer-related genomic rearrangement (GR) or single-nucleotide variant (SNV) at current statistical power, samples with *CHD1*, *RB1* or *NKX3-1* deletions had shorter tumour TL (Fig. 1d). By contrast, TL was closely associated with multiple measures of genomic instability. Tumours with shorter TLs had an elevated number of SNVs ($\rho = -0.27$, $P = 5.78 \times 10^{-8}$; Fig. 2a), indels ($\rho = -0.32$, $P = 2.83 \times 10^{-10}$; Fig. 2c) and GRs ($\rho = -0.12$, $P = 1.63 \times 10^{-2}$; Fig. 2e), as well as higher PGA ($\rho = -0.21$, $P = 3.95 \times 10^{-5}$; Fig. 2g), suggesting tumours with shorter telomeres accrue more mutations of all types without strong selective pressures for specific ones.

To determine whether these associations with somatic features were also related to an individual's non-tumour cells, we related each somatic feature against the TL ratio (tumour TL / non-tumour TL). Similar to tumour TL, the TL ratio did not significantly differ between samples with any of the recurrent prostate cancer-related GRs or CNAs but samples with a somatic SNV in the gene *SPOP* had smaller TL ratios (Supplementary Fig. 2). We identified significant correlations between somatic genomic instability measures and TL ratio. Tumours with an elevated number of SNVs ($\rho = -0.15$, $P = 4.20 \times 10^{-3}$; Fig. 2b), indels ($\rho = -0.18$, $P = 2.97 \times 10^{-4}$; Fig. 2d), GRs ($\rho = -0.22$, $P = 1.08 \times 10^{-5}$; Fig. 2f) and PGA ($\rho = -0.13$, $P = 1.69 \times 10^{-2}$; Fig. 2h) had smaller TL ratios.

We also assessed the association of telomere length with chromothripsis using published ShatterProof[27] scores from a subset of samples in this cohort ($n = 170$)[14]. There was no correlation between scores representing chromothripsis events in

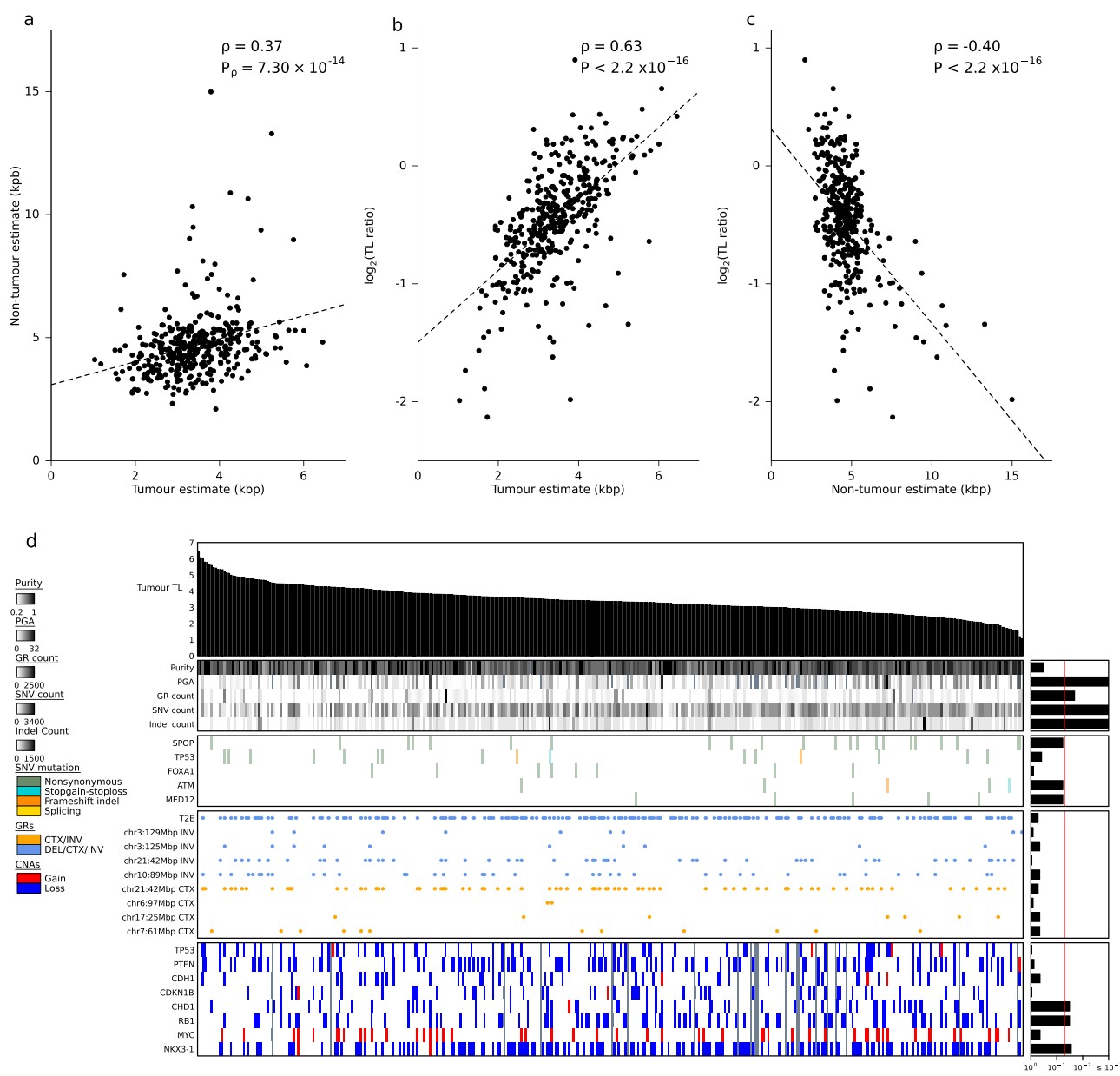

**Fig. 1 Tumour telomere length (TL) is associated with genomic features. a** Spearman correlation between tumour TL and non-tumour TL. **b** Spearman correlation between tumour TL and TL ratio (tumour TL / non-tumour TL). **c** Spearman correlation between non-tumour TL and TL ratio. **d** Tumour TL is ranked in descending order of length (kbp). The association of tumour TL and measures of mutational burden (PGA; percent genome altered), TMPRSS2:ERG (T2E) fusion status, as well as known prostate cancer genes with recurrent copy number aberrations (CNAs), coding single-nucleotide variants (SNVs), indels (insertion and/or deletion) and genomic rearrangements (GRs) are shown. Bar plots indicate the statistical significance of each association (Methods).

either tumour TL ($\rho = 0.06$, $P = 0.43$) or TL ratio ($\rho = 0.02$, $P = 0.80$).

**Fusion events are associated with telomere length**. When telomeres shorten beyond a certain length, double-strand break repair is activated and cell cycle progression is arrested *via* the *TP53* pathway[28]. Failure to block cell growth can lead to telomere crisis and subsequent translocations, chromothripsis or chromosome fusions[29]. We explored the association of TL and the number of gene fusions present in a tumour. There was a negative correlation between the number of gene fusions and tumour TL

($\rho = -0.26$; $P = 2.18 \times 10^{-3}$) but no correlation with TL ratio (Fig. 2i-j). In a previous study, 47 recurrent gene fusions were discovered from matched RNA-Sequencing data[18]. Differences in tumour TL and TL ratio between samples with a gene fusion and those without were investigated for each of these recurrent fusions. No gene fusions were associated with TL ratio, but the PCAT1:CASC21 gene fusion was significantly associated with tumour TL (two-sided Mann–Whitney $U$ test; $n = 139$, $Q = 2.07 \times 10^{-4}$; Supplementary Fig. 3 and Supplementary Data 2). Tumours with this fusion had shorter tumour telomeres (mean = 3.3 kbp) than those without (mean = 3.8 kbp). These

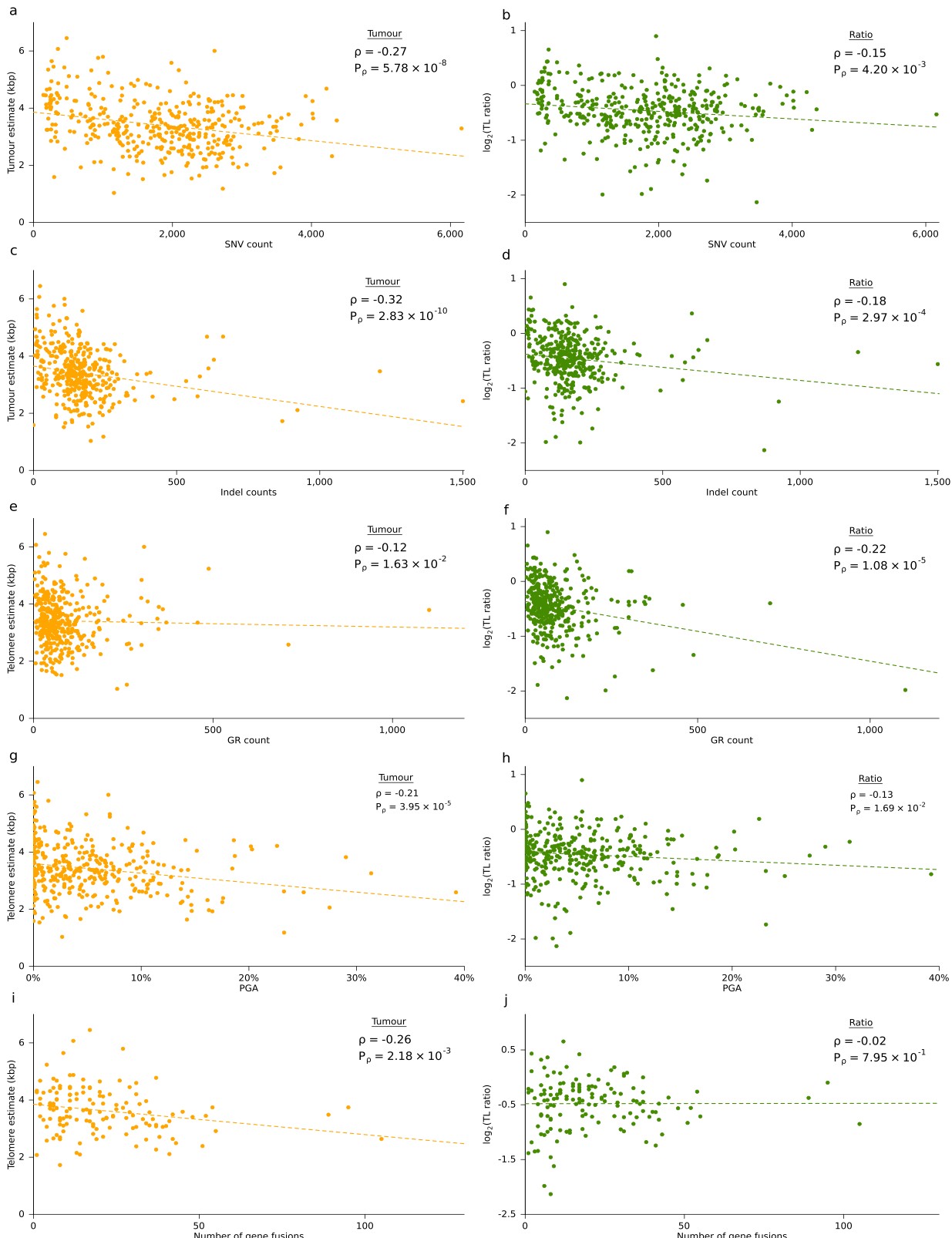

**Fig. 2 Mutational landscape differs with telomere length. a**, **b** Correlation between the number of single-nucleotide variants (SNVs) and **a** tumour telomere length (TL) **b** TL ratio. **c**, **d** Correlation between the number of indels and **c** tumour TL **d** TL ratio. **e**, **f** Correlation between the number of genomic rearrangements (GRs) and **e** tumour TL **f** TL ratio. **g**, **h** Correlation of percentage of the genome altered (PGA) and **g** tumour TL **h** TL ratio. **i**, **j** Correlation between the number of gene fusions and **I** tumour TL **j** TL ratio. Spearman's $\rho$ and $P$-values are displayed, two-sided.

data suggest that the number of fusions and specifically the long non-coding RNA *PCAT1*, which promotes cell proliferation, is related to tumour TL.

**Proliferation rate is not associated to telomere length**. The rapid reproduction or proliferation of a cell should reduce the telomere length in dividing tumour cells. To test this, we investigated the correlation of TL with MKI67 abundance levels and a previously published proliferation score[30]. Surprisingly, there was no association between either tumour TL ($\rho = -0.14$; $P = 0.11$) or TL ratio ($\rho = -0.09$; $P = 0.30$) and MIK67 RNA abundance ($n = 139$, Supplementary Figs. 3b, c). Similarly, there was no association between proliferation scores and tumour TL ($\rho = 0.01$; $P = 0.91$) or TL ratio ($\rho = -0.05$; $P = 0.54$; Supplementary Figs. 3d-e). This suggests that there is a more complex relationship between proliferation and TL at play.

**The role of *TERT* in prostate cancer**. A pan-cancer study reported that *TERT* alterations including promoter mutations, amplifications and structural variants were seen in ~30% of all cancers[7]. In our cohort, 10% of samples had *TERT* amplifications, 11% had *TERC* amplifications, ~1% had *TERT* structural variants and no samples had *TERT* SNVs or gene fusions. *TERT* mutations were seen less frequently in other localized prostate cancer datasets, 1.7% (17/1,013;[31] and 0.6% (2/333;[13]), and in a metastatic dataset 3% (5/150;[20]), likely reflecting the early-stage of our cohort. Mutations in *ATRX* and *DAXX*, which have been correlated with longer telomeres[32], were rare in our cohort: only two samples harboured a CNA in *DAXX*, and only four samples had an alteration in *ATRX*.

Tumour *TERT* RNA abundance was not correlated with tumour TL or TL ratio (Fig. 3a). Samples with higher *TERT* RNA abundance had fewer GRs ($\rho = -0.17$; $P = 4.79 \times 10^{-2}$; Fig. 3b), but there was no correlation between *TERT* abundance and SNV count ($\rho = -0.04$, $P = 0.67$; Fig. 3c), indel count ($\rho = -0.04$, $P = 0.13$; Fig. 3d) or PGA ($\rho = -0.13$, $P = 0.68$; Fig. 3e). The abundance of *TERC*, the telomerase RNA component, was negatively correlated with tumour TL ($\rho = -0.24$; $P = 4.55 \times 10^{-3}$; Supplementary Fig. 4a) but there was no correlation with TL ratio or GR count ($\rho = 0.12$; $P = 0.15$; Supplementary Fig. 4b). *TERC* abundance was positively correlated with SNV count ($\rho = 0.23$; $P = 7.34 \times 10^{-3}$; Supplementary Fig. 4c), indel count ($\rho = 0.34$; $P = 4.88 \times 10^{-5}$; Supplementary Fig. 4d) and PGA ($\rho = 0.26$; $P = 1.90 \times 10^{-3}$; Supplementary Fig. 4e). *TERT* and *TERC* abundances were not correlated ($\rho = 0.02$; $P = 0.794$). These data suggest that TERT signalling is not significantly abrogated in localized prostate cancer either by somatic aberrations or through gene expression changes.

To explore the relationship of *TERT* RNA abundance and tumour TL further, we considered known activating transcription factors. Transcription of *TERT* can be activated by *MYC* and *SP1* and repressed by *AR*[33]. *MYC* amplifications occurred in 14.5% of our samples (51/351; Fig. 1d), while *SP1* CNAs were rare (3/351). *TERT* and *MYC* mRNA abundance was positively correlated ($\rho = 0.27$; $P = 1.46 \times 10^{-3}$) but *MYC* abundance was unrelated to tumour TL (Supplementary Fig. 5a). Contrastingly, there was a positive correlation between tumour TL length and *SP1* abundance ($\rho = 0.23$; $P = 6.84 \times 10^{-3}$) but no significant correlation between *SP1* and *TERT* abundance (Supplementary Fig. 5b). We did not observe any statistically significant correlations between *AR* and *TERT* abundance, or tumour TL (Supplementary Fig. 5c). The direct relationship of these transcription factors on *TERT* is hard to elucidate because of the low measured abundance

of *TERT*. Nonetheless, the abundance of *SP1* and *AR* appear to positively and negatively affect tumour TL, respectively.

To determine whether *TERT* was being regulated epigenetically, we first investigated the correlation between its methylation status and RNA abundance using 91 annotated sites. We identified one CpG site with a significant negative correlation and two with significant positive correlations (two-sided Spearman's correlation; $Q < 0.05$; $|\rho| > 0.2$; Fig. 3f). Further, 31% (28/91) of *TERT* CpGs sites were significantly correlated to tumour TL: seven positively and 21 negatively (two-sided Spearman's correlation; $Q < 0.05$; $|\rho| > 0.2$; Fig. 3f). This strongly suggests that methylation of *TERT* may impact *TERT* abundance and tumour TL.

**Candidate regulators of prostate tumour telomere length**. Evidence of correlation between methylation and tumour TL in *TERT* led us to investigate the role of methylation on TL genome-wide. For each gene, we considered the CpG site most associated to its mRNA abundance (see Methods) and related that to tumour TL ($n = 241$). Methylation of almost half of all genes (46%; 7,088/15,492) was significantly correlated with tumour TL (two-sided Spearman's correlation; $Q < 0.05$; Supplementary Data 3). Similarly, almost a third of genes showed transcriptional profiles associated with tumour TL (32%; 4,520/13,956; two-sided Spearman's correlation; $Q < 0.05$). No proteins were significantly associated with tumour TL after FDR adjustment although 9.3% proteins showed correlation to tumour TL before adjustment ($n = 548/5,881$; two-sided Spearman's correlation; unadjusted $P < 0.05$). There were 112 genes with methylation, transcription and proteome correlations to tumour TL. Remarkably, these showed no functional enrichment. Several genes showed methylation positively correlated with tumour TL but negatively correlated with RNA and protein abundance (Fig. 4a), suggesting suppression of tumour TL elongation. One such gene is the oncogene *AKT1*, which regulates processes including cell proliferation, survival and growth[34]. High *AKT1* abundance may indicate an elevated proliferation and, therefore, shorter telomeres.

We also identified genes whose methylation was negatively correlated with tumour TL but positively correlated with RNA and protein abundance suggesting promotion of telomere elongation (Fig. 4b). These included *SLC14A1*, a membrane transporter that mediates urea transport, and *ITGA3*, an integrin that functions as a cell surface adhesion molecule. We used gprofiler2[35] to identify pathways enriched in genes with methylation or transcriptomic profiles that are correlated with tumour TL using KEGG pathways[36]. We identify 16 pathways enriched in genes with methylation profiles and 16 pathways that were enriched in genes with transcriptomic profiles that were correlated with tumour TL (Supplementary Fig. 6a). To reduce false positives and account for crosstalk between pathways, we applied a crosstalk correction method[37,38]. The crosstalk matrices (Supplementary Figs. 6b, c) identified overlap between the cancer-related pathways, and after crosstalk adjustment only one pathway remained enriched in genes with transcriptomic profiles that were correlated to tumour TL (hsa04519, focal adhesion; Supplementary Figs. 6d, e).

We similarly investigated whether TL ratio was associated with methylation and found that the methylation levels of 33.7% (5,218/15,492) of genes were significantly correlated with TL ratio (two-sided Spearman's correlation; $Q < 0.05$; Supplementary Data 4). Surprisingly, <1% ($n = 53/13,958$) of genes with overlapping data also had a significant correlation between RNA abundance and TL ratio and none between protein

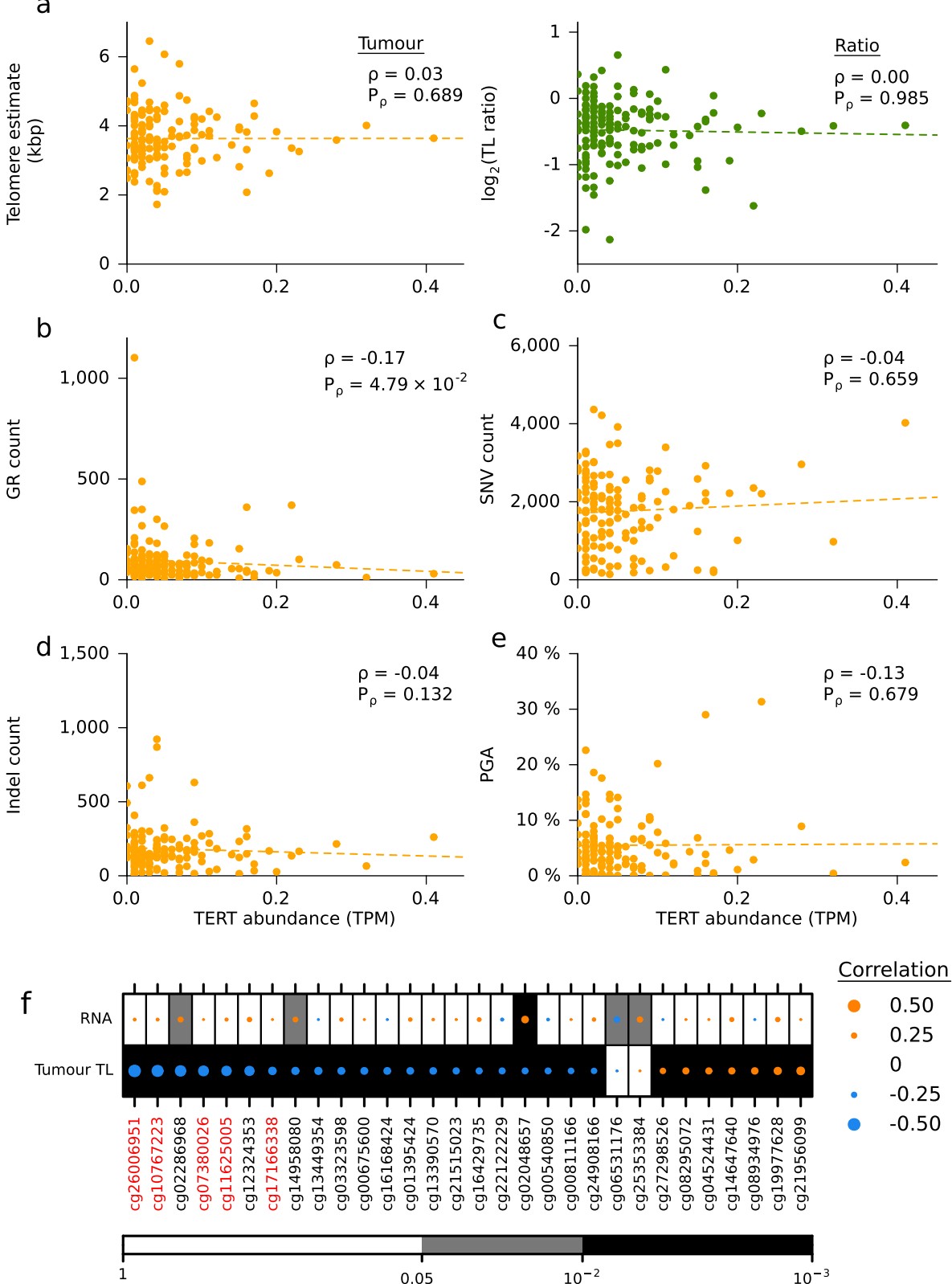

**Fig. 3 The genomic correlates of *TERT* abundance. a** Correlation of *TERT* RNA abundance with tumour telomere length (TL) and TL ratio. Spearman's $\rho$ and *P*-values are displayed, two-sides. **b**–**e** Correlation of *TERT* abundance and **b** the number of GRs, **c** number of SNVs, **d** number of indels and **e** PGA. Spearman's $\rho$ and *P*-values are displayed, two-sided. **f** Spearman's correlation of significantly associated methylation probes with RNA abundance and tumour TL. Probes within the promoter are labelled in red while the rest are located in the gene body. Dot size indicated the magnitude of correlation. Background colour indicates unadjusted *P*-values. Methylation probes are ordered by their correlation between *TERT* RNA abundance from negative to positive.

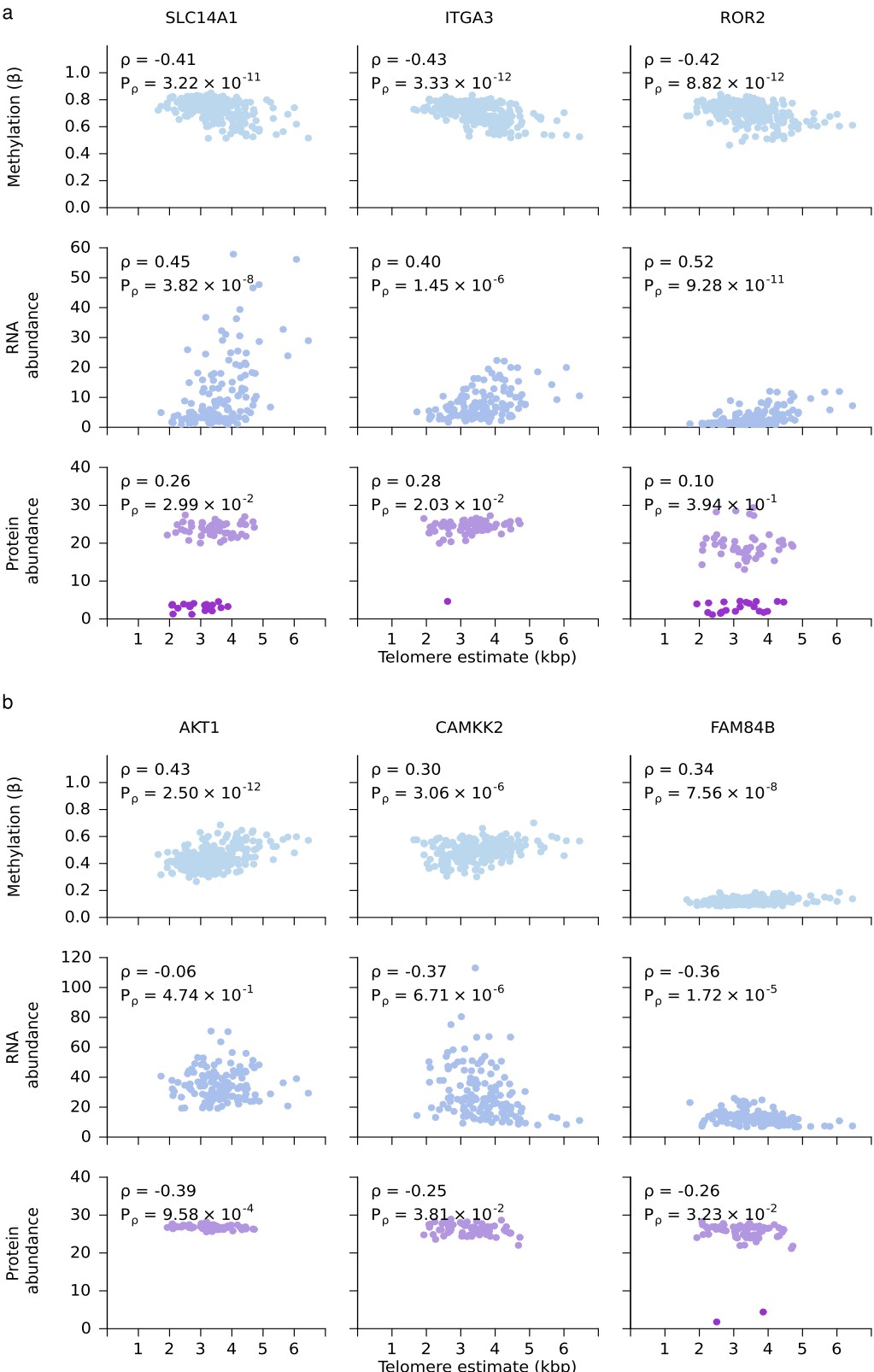

**Fig. 4 Association of methylation, RNA abundance, protein abundance and TL. a** Positive correlation of methylation and tumour telomere length (TL), but negative correlation of RNA and protein abundance. **b** Negative correlation of methylation and tumour TL, but positive correlation of RNA and protein abundance. Darker purple dots represent undetected, imputed protein abundance measures. Spearman's $\rho$ and *P*-values are displayed, two-sided.

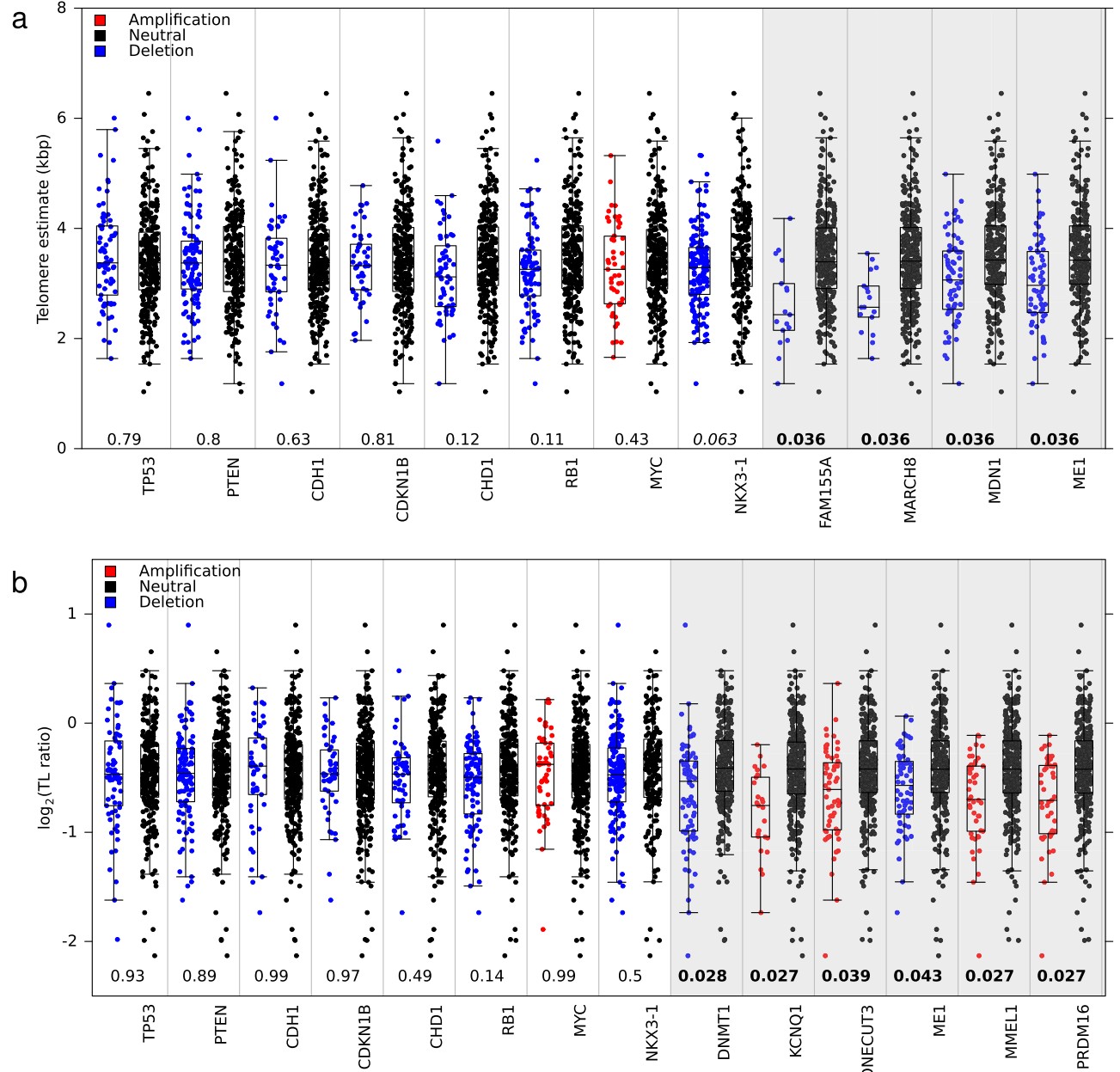

**Fig. 5 Telomere length differs by copy number status. a, b** Difference in **a** tumour TL and **b** TL ratio between samples ($n = 381$) with a copy number aberration and those without in prostate cancer-related genes and associated genes. Q values are from a two-sided Mann–Whitney U test and are bolded when significant (FDR < 0.05). Box plots depict the upper and lower quartiles, with the median shown as a solid line; whiskers indicate 1.5 times the interquartile range (IQR).

abundance and TL ratio (two-sided Spearman's correlation; unadjusted $P < 0.05$). These results suggest that tumour TL, not TL ratio, is associated with tumour gene expression.

**Association of telomere length and specific CNAs.** Since prostate tumour gene expression and clinical behaviour is predominantly driven by CNAs[14,19] we next investigated their role in TL. As noted above (Fig. 1d), driver CNAs were largely unassociated with tumour TL (Fig. 5a; white background) or TL ratio (Fig. 5b; white background). We, therefore, considered copy number changes genome-wide for associations with TL. We identified 24 loci encompassing 35 genes in which there was a significant difference in tumour TL in samples with a copy number change compared to those without (two-sided Mann–Whitney U test; $Q < 0.05$; Supplementary Data 5

and Fig. 5a). We also identified 128 loci encompassing 319 genes in which there was an association between copy number status and TL ratio (two-sided Mann–Whitney U test, $Q < 0.05$; Supplementary Data 6). For example, tumours with deletions in DNA methyl-transferase 1, *DNMT1*, had smaller TL ratios ($Q = 0.028$, effect size = 0.11, Fig. 5b). An opposing trend was seen in the chromatin organization gene, *PRDM16* ($Q = 0.027$, effect size = 0.15) and the membrane metallo-endopeptidase gene, *MMEL1* ($Q = 0.027$, effect size = 0.14; Fig. 5b), where amplifications resulted in smaller TL ratios. This analysis highlights that copy number aberrations are more associated with TL ratio (change in length from non-tumour TL to tumour TL) than absolute tumour TL.

We also explored CNAs in genes comprising the telomere complex (*TERF1, TERF2, TERF2IP* and *POT1*), shelterin inter-acting proteins (*PINX1* and *RTEL1*), and the components of

telomerase (*TERT* and *TERC*). There were no differences in the tumour TL (Supplementary Fig. 7a) or TL ratio (Supplementary Fig. 7b) between samples with and without a CNA in these genes.

Next, we compared TL across previously identified CNA subtypes. There was no difference in tumour TL ($P = 0.53$; one-way ANOVA) or TL ratio ($P = 0.78$; one-way ANOVA) in the four CNA subtypes identified from aCGH arrays and associated with prognosis[9] (Supplementary Fig. 8a, b). There was an association between TL ratio and the six CNA subtypes ($P = 3.13 \times 10^{-2}$; one-way ANOVA) identified from 284 OncoScan SNP arrays[14] but not with tumour TL (Supplementary Fig. 8c, d). Samples in subtype C5, which was defined by amplifications in genes near the end of chromosomes had smaller TL ratios than C3 (defined by an 8p deletion and an 8q amplification) and C4 (defined as having a quiet CNA profile). A smaller TL ratio in the samples from subtype C5 indicates that the non-tumour TL length was longer than in the tumour TL (Supplementary Fig. 8e): the consequences of this remain to be elucidated.

**Clinical correlates of telomere length.** The clinical features of a tumour can have prognostic value, and have been associated with the genomic features of tumours[14]. Higher serum abundance of prostate specific antigen (PSA), higher ISUP Grading and tumour size and extent are all associated with worse outcome. Therefore, we considered whether there was interplay between TL and the clinical features of a tumour. Tumour TL was not significantly correlated to age at diagnosis, ($\rho = -0.10$, $P = 5.80 \times 10^{-2}$; Fig. 6a) but there was a significant positive correlation between age and TL ratio ($\rho = 0.11$, $P = 2.53 \times 10^{-2}$; Fig. 6b). Tumour TL was shorter than non-tumour TL in younger patients. This could be related to the aggressiveness of early onset prostate cancers, which is characteristic of tumours in younger men[24]. There was a negative correlation between pre-treatment PSA levels between both tumour TL ($\rho = -0.16$, $P = 2.23 \times 10^{-3}$) and TL ratio ($\rho = -0.19$, $P = 1.70 \times 10^{-4}$; Fig. 6c, d). Neither tumour TL nor TL ratio was associated with ISUP Grade (Fig. 6e, f). Surprisingly, tumour TL was shorter in smaller tumours (T1) than larger tumours (T2 or T3; one-way ANOVA, $P = 2.2 \times 10^{-2}$; Fig. 6g) but this can be explained by the higher average age of patients with T1 tumours (mean = 71.3 years) compared to other T categories (mean = 62.0 years). Accordingly, there was no association between TL ratio, which controls for patient age, and T category ($P = 0.29$; Fig. 6h).

Telomerase activity and TL has been proposed to have clinical utility at three different stages; diagnosis, prognosis and treatment[33]. TL from biopsies has been correlated with progression to metastasis and disease specific death[39]. As well, TL from leucocytes has been associated with poor survival[40,41]. We explored if tumour TL, non-tumour TL or TL ratio were associated with biochemical relapse (BCR), an early surrogate endpoint in intermediate-risk prostate cancer. Cox proportional hazards (Cox PH) models were fit, splitting patients ($n = 290$) into two groups based on their TL with increasing cutoff thresholds using the group with the longer TL as the reference group. (50 bp each time; Supplementary Figs. 9a–c). From this outcome-oriented optimal cut point analysis we discovered that samples with non-tumour TL > 3.9 kbp had a lower rate of BCR than samples with shorter TLs (HR = 2.02, $P = 1.6 \times 10^{-3}$; Fig. 6i). Non-tumour TL is associated with survival independent of PGA (Cox PH model, $P = 0.02$). There was no association between tumour TL and BCR (Fig. 6j), but there was an association between TL ratio and BCR, where samples with a TL ratio > −0.6 had a lower rate of BCR (HR = 0.42, $P = 2.6 \times 10^{-3}$; Fig. 6k). We also considered TL as a continuous measurement

and fit Cox PH models using tumour TL, non-tumour TL and TL ratio. Again, there was no association between continuous tumour TL and BCR but there was an association between non-tumour TL (HR = 0.77, $P = 1.4 \times 10^{-2}$) and TL ratio (HR = 1.71, $P = 3.1 \times 10^{-2}$; Supplementary Fig. 9d). These results suggest that non-tumour TL and TL ratio are weakly prognostic, and thus may reflect host factors that may influence patient risk categorization.

## Discussion
These data emphasize the relationship of genomic instability and TL. Genomic instability has previously been linked with poor outcome in prostate cancer[9,14] and TL shortening could be the cause of some of this instability. Telomere shortening has been implicated as an early event in prostate cancer due to evidence of shortened telomeres observed in a precursor histopathology, high-grade prostatic intraepithelial neoplasia[42,43]. Since cellular proliferation in prostate cancer is increased by sevenfold compared to normal prostatic epithelial cells[33], telomeres in these dividing cells will shorten with each cell division. There is no evidence that primary prostate cancer exhibits ALT lengthening[23] therefore the vast majority, if not all tumours, activate telomerase for telomere maintenance. We did not observe any *TERT* promoter mutations in our cohort but there are strong negative correlations between methylation probes in the promoter of *TERT* and tumour TL. This may be a proxy for telomerase activity since DNA methylation impedes transcription.

We see an unexpected divergence between somatic molecular features associated with TL ratio and those with tumour TL. Specifically, measures of genomic instability are linked to TL ratio (which represents the ratio between tumour TL and non-tumour TL) while specific CNAs, GRs and SNVs are not (Fig. 1 and Supplementary Fig. 2). This suggests that during the progression of cells from normal to cancerous, non-tumour TL may influence tumour genomics, where tumours with shorter TL experience more genomic instability. Alternatively, a common factor may be influencing during this epoch of the tumour's evolution. Once tumours are formed, it is the specific mutations within the cell that are more associated with tumour TL. This may be due to mutations in cell division and growth-regulating genes such as *ATK1* and *SPOP*, which increases the number of divisions in the tumour and thereby shortens tumour telomeres. Further evidence of this hypothesis is seen in tumours with *PCAT1* fusions, where tumours with this fusion had shorter tumour TL than samples without it[44].

One limitation in the estimation of TL using short-read whole genome sequencing is the difficulty in estimating chromosome specific telomere lengths. Junction spanning reads from paired-end experiments, in which one read maps within the first or last band of the chromosome and the other read maps within the telomere region, are scarce. Further studies should be performed using-long read sequences, in which these regions may have more coverage and can be used to determine chromosome specific shortening and its association to specific genomic events or biochemical relapse.

These data highlight the complicated relationship between telomere length in both tumour and non-tumour cells, and molecular and clinical tumour phenotypes. They highlight the need for increased study of telomere length across cancer types, and for long-read sequencing to introduce chromosome-specific analyses.

## Methods
**Patient cohort.** Published whole genome sequences of tumour and matched non-tumour samples, as well as, available clinical information were downloaded from public repositories (phs000447.v1.p1[11], phs000330.v1.p1[12], EGAS00001000900[14],

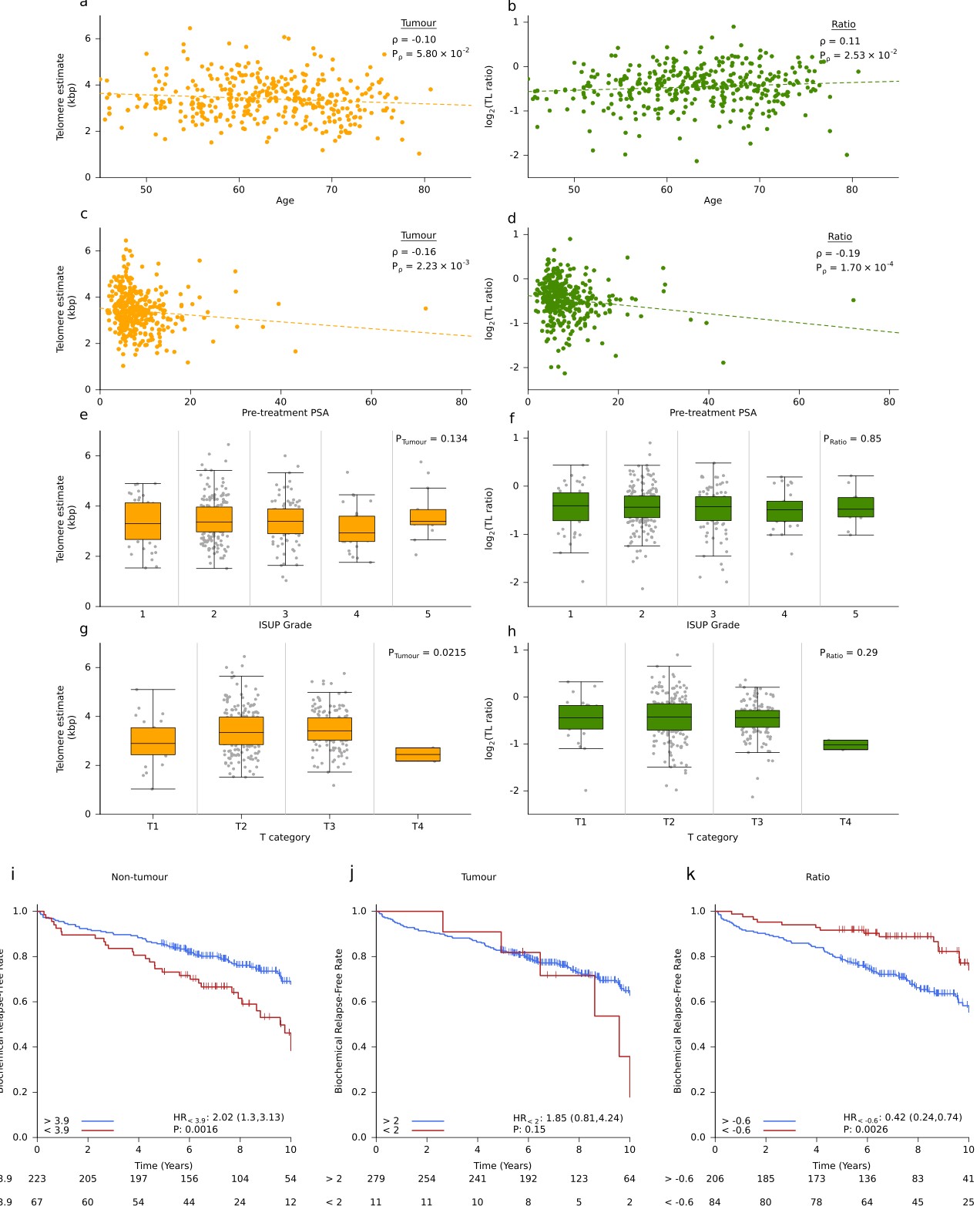

**Fig. 6 Telomere length is associated with clinical features and biochemical relapse. a**, **b** Correlation of age at diagnosis with **a** tumour telomere length (TL) and **b** TL ratio. Spearman's $\rho$ and *P*-values are displayed, two-sided. **c**, **d** Correlation of pre-treatment prostate specific antigen (PSA) with **c** tumour TL and **d** TL ratio. Spearman's $\rho$ and *P*-values are displayed, two-sided. **e**, **f** Association of ISUP (International Society of Urological Pathology) grade with **e** tumour TL and **f** TL ratio. *P*-value is from a one-way ANOVA, $n = 381$. Box plots depict the upper and lower quartiles, with the median shown as a solid line; whiskers indicate 1.5 times the interquartile range (IQR). **g**, **h** Association of T category with **g** tumour TL and **h** TL ratio. *P*-value is from a one-way ANOVA, $n = 381$. Box plots depict the upper and lower quartiles, with the median shown as a solid line; whiskers indicate 1.5 times the interquartile range (IQR). On all plots, green indicates TL ratio, while orange indicates tumour TL. **i**, **k** Cox proportional hazard models were created for **i** non-tumour TL, **j** tumour TL and **k** TL ratio with biochemical relapse (BCR) as the endpoint. Samples ($n = 290$) were split into two groups based on the optimal cut point analysis (Methods).

phs000178.v11.p8[13], EGAS00001000400[24], phs001648.v2.p1[45]). Cellularity was also determined in silico from OncoScan SNP arrays via qpure (v1.1)[46]. Detailed clinical characteristics of patients are provided in Supplementary Data 1. Informed consent, consistent with the guidelines of the local Research Ethics Board (REB) and International Cancer Genome Consortium (ICGC), was obtained at the time of clinical follow-up. Previously collected tumour tissues were used, following University Health Network REB-approved study protocols (UHN 06-0822-CE, UHN 11-0024-CE, CHUQ 2012-913:H12-03-192) and IRB #21-009599. Participants were not compensated for their involvement.

**Whole genome sequencing data analysis.** Raw sequencing reads were aligned to the human reference genome, GRCh37, using BWA-mem (>v0.7.12;[47]) at the lane level. Picard (v1.92; http://broadinstitute.github.io/picard/) was used to merge the lane-level BAMs from the same library and mark duplicates. Library level BAMs from each sample were also merged without marking duplicates using Picard. Local realignment and base quality recalibration was carried out on tumour/non-tumour pairs together using the Genome Analysis Toolkit (GATK; > v3.4.0;[48]). Tumour and non-tumour sample level BAMs were extracted, headers were corrected using SAMtools (v0.1.9;[49]), and files were indexed with Picard.

**Computational telomere length estimation.** Tumour and non-tumour telomere lengths were estimated using TelSeq (v0.0.1;[25]) and TelomereHunter (v1.0.4)[26] on BAM files generated using BWA-mem (>v0.712;[47]) and GATK (>v3.4.0;[48]). TelSeq estimates telomere length with the following formula $l = t_k sc$, where $t_k$ is the abundance of telomeric reads (reads that contain $k$ or more TTAGGG repeats; $k = 7$), $c$ is a constant for the genome length divided by the number of chromosome ends and $s$ is the fraction of all reads with GC composition between 48–52%. TelomereHunter also identifies telomeric reads based on the number to repeat sequences with a read and normalizes by the number of reads with GC content between 48–52%. This value is multiplied by $10^6$ to calculate TRPM (telomeric reads per GC content-matched million reads) values. As a quality measure, TelSeq estimates for each sample were generated per sequencing lane. Reads from lanes that contained too few reads to calculate an estimate (marked as UNKNOWN), and outlier lanes as identified by grub's test, were removed from input BAMs using BAMQL (v1.6)[50]. After outliers were removed, TelSeq was run again ignoring read groups with the $-u$ parameter. Samples with telomere estimates <0.25 were removed from further analysis. To account for differences in TL due to sequencing centre, a linear model was fit with TL as the response variable and sequencing centre as the predictor variable. A separate model was fit for tumour and non-tumour length.

**Somatic variant calling.** Single-nucleotide variants (SNVs) and genomic rearrangements (GRs) were called using standardized pipelines across all datasets. SomaticSniper (v1.0.5;[51]) was used to call SNVs on bases with at least 17x coverage in tumours and 10x in non-tumours. Coding versus non-coding SNVs were determined using Annovar[52]. Genomic rearrangements were identified using Delly (v0.7.8;[53]). Gene fusion events involving *ERG* or *ETV* were collectively referred to as ETS events. Genomic rearrangement calls were examined to determine if breakpoints led to a TMPRSS2:ERG fusion or if breakpoints were found in both 1 Mbp surrounding the following gene pairs: *ERG:SLC45A3, ERG:NDRG1, ETV1:TMPRSS2, ETV4:TMPRSS2, ETV1:SLC45A3, ETV4:SLC45A3, ETV1:NDRG1* and *ETV4:NDRG1*. *ERG* immunohistochemistry and deletion calls between *TMPRSS2* and *ERG* loci in OncoScan SNP array data provided further support for these fusions.

**mRNA abundance data generation and analysis.** Total RNA was extracted with mirVana miRNA Isolation Kit (Life Technologies) according to manufacturer's instructions. RNA samples were sent to BGI Americas where it underwent QC and DNase treatment. Two hundred nanogram of total RNA was used to construct a TruSeq strand specific library with the Ribo-Zero protocol (Illumina), and all samples were sequenced on a HiSeq2000v3 to a minimal target of 180 million paired-end reads. Reads were mapped using the STAR aligner (v2.5.3a;[54]) to GRCh37 with GENCODE v24lift37[55]. RSEM (v1.2.29) was used to quantify gene abundance[56].

**Methylation microarray data generation.** Illumina Infinium HumanMethylation 450k BeadChip kits were used to assess global methylation, using 500 ng of input genomic DNA, at McGill University and the Genome Quebec Innovation Centre (Montreal, QC). All samples used in this study were processed from fresh-frozen prostate cancer tissue. The IDAT files were loaded and converted to raw intensity values with the use of wateRmelon package (v1.15.1;[57]). Quality control was conducted using the minfi package (v1.22.1;[58] no outlier samples were detected). Raw methylation intensity levels were then pre-processed using Dasen. For each probe, a detection *P*-value was computed to indicate whether the signal for the corresponding genomic position was distinguishable from the background noise. Probes having 1% of samples with a detection $P < 0.05$ were removed. We also filtered probes based on SNPs and non-CpG methylation probes. Next, we used the

DMRcate package (v1.4.2) to further filter out 27,309 probes that are known to cross-hybridize to multiple locations in the genome and 17,168 probes that contain an SNP with an annotated minor allele frequency of >5% with a maximum distance of two nucleotides to the nearest CpG site. Annotation to chromosome location, probe position, and gene symbol was conducted using the IlluminaHumanMethylation450kanno.ilmn12.hg19 package (v0.6.0).

**Association of telomere length with gene fusions.** The association between gene fusion status and tumour TL and TL ratio ($n = 139$) was tested using a two-sided Mann–Whitney *U* test in 47 previously identified[18] recurrent gene fusions.

**Association of telomere length and proliferation.** A proliferation score per sample was generated using a previously published signature[30] where tumours with an RNA abundance value greater than the mean for each gene in the signature were given a score of $+1$, and tumours with an RNA abundance value less than the mean for that gene were given a score of $-1$. All values were summed to generate a proliferation score. Two-sided Spearman's correlations between TL, TL ratio and the proliferation score was calculated. The two-sided Spearman's correlation between TL, TL ratio and MKI67 abundance was also calculated.

**Association of telomere length with chromothripsis.** Chromothripsis scores were previously generated using ShatterProof (v0.14;[14,27]) with default settings. Two-sided Spearman's correlation between the maximum ShatterProof score per sample and telomere length was calculated using samples with both available metrics ($n = 170$).

**Association of TL with clinical and genomic features.** Clinical features, including ISUP Grade, pre-treatment PSA, T category and age at diagnosis, were categorized and tested for association using a one-way ANOVA. Pathological T category was used for surgery samples and diagnostic T category was used for radiotherapy samples. Binary features including the presence of specific GRs, CNAs and SNVs were tested for association using a two-sided Mann–Whitney *U* test. Summary features including PGA, GR count, SNV count and indel count were correlated to TL using two-sided Spearman's correlation using the AS-89 algorithm[59] as implemented in the base *R* stats package.

**Association of telomere length with methylation.** The correlation matrix of methylation and mRNA abundance levels from TCGA was downloaded from https://gdac.broadinstitute.org/. For each gene, the probe showing the highest Spearman's correlation with mRNA abundance levels was used in our correlation analysis. Two-sided Spearman's correlations between TL and methylation beta values ($n = 241$) were calculated using the AS-89 algorithm[59] as implemented in the base *R* stats package.

**Association of TL with transcriptome and proteome abundance.** Two-sided Spearman's correlations between TL and RNA ($n = 138$;[14]) and protein abundance ($n = 70$;[17]) and TL were calculated using the AS-89 algorithm[59] as implemented in the base *R* stats package.

**Over-representation analysis pathway analysis.** Pathway analysis was performed with the gprofiler2[35] (v0.2.1) *R* package using genes in which there was a significant association between TL and methylation or RNA separately using KEGG collection of pathways[36].

**Crosstalk effects in pathway analysis.** To account for crosstalk effects caused by gene overlap in pathway analysis, we implemented the principle component analysis method proposed by[38]. Briefly, for genes that are overlap among pathways, each gene is only allowed membership in one of the pathways. This membership is determined by the highest correlation between the gene and the PC1 of the other genes in the pathway. An one-sided Fisher's Exact test was then used to determine enrichment of TL correlated genes in the reduced pathway membership.

**Association of telomere length with CNAs.** SNP microarrays were performed with 200 ng of DNA on Affymetrix OncoScan FFPE Express 2.0 and 3.0 arrays. Analysis of the probe assays was performed using.OSCHP files generated by OncoScan Console (v1.1) using a custom reference. BioDiscovery's Nexus Express™ for OncoScan 3 Software was used to call copy number aberrations using the SNP-FASST2 algorithm. Gene level copy number aberrations for each patient were identified by overlapping copy number segments from OncoScan SNP 3.0 data, with RefGene (2014-07-15) annotation using BEDTools (v2.17.0;[60]). Genes with the same copy number profile across patients were then collapsed into contiguous regions. Contiguous gene segments with aberrations in <5% of patients were removed from the analysis. To find associations between TL and copy number segments, a two-sided Mann–Whitney *U* test was used to compare the mean TL between samples with a copy number aberration and those without ($n = 381$). The copy number aberration state (either amplified or deleted) was determined as the

status with the largest proportion of samples. Samples with aberrations in the other class was merged into the without group. For example, three samples have an amplification in *CHD1*, while 49 samples have a deletion. The three samples would be grouped with copy number neutral samples and the Mann–Whitney test performed comparing the two groups. *P*-values were FDR adjusted to account for multiple testing.

**Association with biochemical relapse**. Cox proportional hazards models were fit with the *R* package survival (v3.2–7) using TL as a continuous or discrete variable. Age at diagnosis was controlled for in the model. Kaplan Meier plots were generated by dichotomizing samples based on the optimal cut point analysis, in which samples were dichotomized using increasing thresholds of 50 bp.

**Statistical analyses and data visualization**. All statistical analyses were performed within the *R* statistical environment (v4.1.1). Visualization in *R* was performed through the BoutrosLab Plotting General package (v5.6.1;[61]). We chose to use Spearman's rank correlation because it is a nonparametric test and does not assume that values are sampled from a population that follows a Gaussian distribution or that there is a linear relationship between the variables. *P*-values from two-sided Spearman's correlations were calculated using the AS-89 algorithm[59] as implemented in the base R stats package. Box plots depict the upper and lower quartiles, with the median shown as a solid line; whiskers indicate 1.5 times the interquartile range (IQR).

**Reporting summary**. Further information on research design is available in the Nature Research Reporting Summary linked to this article.

## Data availability

The publicly available WGS data used in this study are available in the EGA and dbGaP databases under the following accession codes; Baca: phs000447.v1.p1[11] [https://www.ncbi.nlm.nih.gov/projects/gap/cgi-bin/study.cgi?study_id=phs000447.v1.p1], Berger: phs000330.v1.p1[12], ICGC PRAD-CA: EGAS00001000900[14], TCGA PRAD: phs000178.v11.p8[13], EO-PCA: EGAS00001000400[24], WCDT-MCRPC: phs001648.v2.p1[45]. These data are available under controlled access after authorization by a Data Access Committee. Access can be requested via EGA or dbGaP. Publicly available processed variant calls (CNAs, GRs, SNVs, indels) are available through the ICGC Data Portal under the project PRAD-CA (https://dcc.icgc.org/projects/PRAD-CA). Publicly available OncoScan SNP array data and RNA-Sequencing data used in this study can be found on EGA under the accession EGAS00001000900. The publicly available mRNA data used in this study are available in the Gene Expression Omnibus (GEO) database under the accession GSE84043. The publicly available methylation data used in this study are available in the GEO database under the accession GSE107298. The publicly available processed proteomics data are available in supplementary material online [https://doi.org/10.1016/j.ccell.2019.02.005]. The publicly available processed RNA-Seq data and gene fusion data are available in supplementary material online [https://doi.org/10.1016/j.cell.2019.01.025]. The data generated in this study, including tumour and non-tumour telomere lengths and association statistics, are available within the article or Supplementary Information.

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

## Acknowledgements

The authors thank all members of the Boutros lab for insightful commentary and technical support. This study was conducted with the support of the Ontario Institute for Cancer Research to P.C.B. through funding provided by the Government of Ontario and by CIHR operating grant #388344. This work was supported by Prostate Cancer Canada and is proudly funded by the Movember Foundation—Grant #RS2014-01. Dr Boutros was supported by a Terry Fox Research Institute New Investigator Award and a CIHR New Investigator Award. This work was funded by the Government of Canada through Genome Canada and the Ontario Genomics Institute (OGI-125). This work was supported by the NIH/NCI under award number P30CA016042, by an operating grant from the National Cancer Institute Early Detection Research Network (U01CA214194) and by support from the ITCR (U24CA248265).

## Author contributions

Formal analysis: J.L. Methodology: J.L., S.E. and E.D. Data curation: J.L., Y.S., T.N.Y., L.E.H., V.H., R.L. T.G. and B.C. Visualization: J.L., J.G. Supervision: M.F., T.v.d.K., R.G.B. and P.C.B. Conceptualization, Supervision: P.C.B., M.F. and R.G.B. Pathology Reviews: T.v.d.K. Writing—original draft: J.L., P.C.B. Writing—review & editing: J.L., T.N.Y., V.H., R.L., M.F. and P.C.B. Approved the manuscript: All authors.

## Competing interests

The authors declare no competing interests.

## Additional information

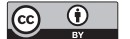

