## [Peer Review File · Nature Communications]

The Telomere Length Landscape of Prostate CancerReviewers' Comments:

Reviewer #1:

Remarks to the Author:

The telomere length landscape of localised prostate cancer
by Livingstone et al.

This is an interesting study that leverages a large body of data generated on this cohort and described previously in multiple publications. Given the highly variable course of prostate cancer (including intermediate risk that this series forces on), linking molecular features to outcomes is valuable. In my opinion the research questions mentioned in the introductions are answered.

General points

Although, the TL ratio is a value corrected for age, the association to age is quite low ($\rho = 0.11$). This result does not support the use of ratio as age-corrected. Also, as far as I know, tissues have its own normal cell division rate. It might be interesting to compare TL of tumour tissues with the one of non-tumour prostate tissue and not only with the one of blood. I'm also unsure how the tumor TL length was calculated from NGS data that has both noncancerous and cancerous prostate epithelial cells. is this tumor or prostate specific?

I also wonder if tumour TL associates with the proliferation rate in general measured by using e.g. a specific gene expression level (as Ki67 ?) although they mention an association with gene fusions that increase cell proliferation. The authors focus mainly on the abundance of TERT and factors that influence TERT expression.

The reported significant correlations are very low. Alos, it is actually Spearman's ρ not Spearman ρ

In some cases the p-value is reported, it is unclear if it was adjusted for multiple testing (only reported for CNAs, Methylation, and TERT abundance). In the supplement tables, some columns mention q-values others p-value. P is always italicized.

How was the TL length estimated: as sum or mean of reads over all chromosomes/solely allosomes ?
Is there any chance to report the TL on chromosome level ?

Figure 2 I-J

Should be called gene fusions not just fusions.

Figure 5

The Mann-Whitney test was run deleted vs non-deleted and amplified vs non-amplified. However, the red or blue dots among the black dots are very difficult to see. Maybe use of an alpha parameter can be considered.

Data

The authors analyse data from 392 patients collected from several publications, e.g. from Fraser et al., which reports 200 tumours. I cannot find a list presenting the data source and collected samples/data.

Method section

Whole-genome sequencing and data analysies

Performed process steps are standard, and published previously

Computational telomere length estimation

In the Methods part, only TelSeq for telomere length estimation is mentioned but in the Result section also TelomereHunter was mentioned. The process description is missing in the Method, although there is a tiny description in the legend text of Figure S1. How were the results of both compared and differences considered.

Association of TL length with transcriptome abundance - Pathway annotation

They used g:Profiler but did not mention which databases they considered (KEGG, Reactome etc). Just a tiny additional comment, over-representation analysis (ORA) is a very insensitive method to find enriched pathways. I would always suggest a crosstalk-based method. In the result section, the authors mention no functional enrichment for methylated genes.

Reviewer #2:

Remarks to the Author:

They use telomere repeat content of whole genome sequence reads as a surrogate marker for telomere length (TL). They show no apparent link between TL and known prostate cancer mutations. As expected there was a weak association between shorter telomere length and measures of genome complexity (SNV, Indels, Genomic rearrangement and gene fusions etc) but no association with chromothripsis.

They describe amplifications at the TERT and TERC and that expression levels didn't correlate with TL, neither did MYC levels. They revealed a correlation between SP1 and TL, but not TERT, and a relationship between a CpG and TERT expression. They then associate telomere length with genome wide methylation patterns - with nearly 50% of methylated genes correlated with telomere length. A subset of genes had methylation, transcriptional and proteomic correlations with TL, but no functional enrichment.

They were able to show that short blood telomere length, but not tumour telomere length, was associated with a higher rate of biochemical relapse. Unlike other previous studies, overall survival was not evaluated as an end point.

This is a nice multi-omic study that brings several strands together ie genomic complexity, telomerase expression, methylation patterns and clinical features to provide further evidence for a role in telomere dysfunction and genomic complexity in prostate cancer. However it is a descriptive study and it wasn't clear what key new information is provided, as such it appears to be a somewhat incremental.

I have no major issues with the data as it is presented.

Minor points:

Supplementary figure 9, indicates a U shaped distribution for HRs as a function of telomere length in the tumour samples. What is the significance of this? Is this consistent with previous reports (Svenson et al Tumour Biol. 2017; Renner et al. Prostate Cancer 2018) that long telomere length in blood is associated with poor survival.

State in methods why Spearman's correlation was chosen over Pearson's

Line 114-115 - statement that 'samples with CHD1, RB1 or NKX3-1 deletions had shorter tumour TL (Fig. 1D)' is not apparent from the data shown in fig1 D. They should provide a data plot that illustrates this point or remove this statement.

Summary

We thank the reviewers for their comments and suggestions. We have updated the manuscript and addressed all concerns from the reviewers. In addition, we have added a comparison of the telomere length from the 382 localized prostate cancer samples and 101 metastatic prostate cancer samples. Key updates include adding:

- Comparison of the TL of tumour and blood tissues with that of 40 adjacent histologically normal paired prostate tissues
- Comparison of TL with MKI67 RNA abundance and a published proliferation score to test association of tumour TL with proliferation rate
- Crosstalk-based adjustment method for the pathway enrichment analysis

Reviewer #1

This is an interesting study that leverages a large body of data generated on this cohort and described previously in multiple publications. Given the highly variable course of prostate cancer (including intermediate risk that this series forces on), linking molecular features to outcomes is valuable. In my opinion the research questions mentioned in the introductions are answered.

We thank the reviewer for kind words and appreciate their detailed comments and suggestions. The suggested analyses have substantially improved the manuscript.

General points:

1. Although, the TL ratio is a value corrected for age, the association to age is quite low ($\rho = 0.11$). This result does not support the use of ratio as age-corrected. Also, as far as I know, tissues have its own normal cell division rate. It might be interesting to compare TL of tumour tissues with the one of non-tumour prostate tissue and not only with the one of blood.

We thank the reviewer for this comment and agree that it would be interesting to compare tumour TL with non-tumour prostate tissue. We therefore quantified the telomere length in 40 adjacent, histologically normal prostate tissue samples. There is a difference in the TL of the adjacent normal samples when compared to TLs from blood tissue and tumour tissue. We have added the following to the manuscript:

Adjacent normal TLs ($n = 40$) were longer than those in blood tissue ($n = 341$; $P = 2.80 \times 10^{-10}$; Mann -Whitney U test) and tumour tissue ($P = 3.04 \times 10^{-21}$; Mann Whitney U test; **Supplementary Fig. 1A**).

Supplementary Fig 1A. Comparison of TL in adjacent, histologically normal prostate tissue, blood and tumour tissue. P values are from a Mann Whitney U-test comparing adjacent normal TL to blood (left) and tumour TL (right).

2. I'm also unsure how the tumor TL length was calculated from NGS data that has both noncancerous and cancerous prostate epithelial cells. is this tumor or prostate specific?

We appreciate the reviewer's comment and agree that tumour purity is a known concern when working with DNA sequencing of bulk tumour tissue. We therefore investigated the association of computationally estimated tumour purity and TL, and happily did not find one (**Supplementary Fig. 1B**). This suggests that TL estimates are robust to the range of purity (~50-90%) in the samples analyzed here. Since there is no relationship between tumour purity and TL, we concluded that no further adjustments to our statistical modeling were necessary.

3. I also wonder if tumour TL associates with the proliferation rate in general measured by using e.g. a specific gene expression level (as Ki67 ?) although they mention an association with gene fusions that increase cell proliferation. The authors focus mainly on the abundance of TERT and factors that influence TERT expression.

We thank the reviewer for this comment and agree that this is an interesting question to investigate. To test if there is an association between proliferation and telomere length, we investigated both MKI67 abundance levels and a published proliferation score (Starmans et al., 2012). Surprisingly, there was no significant correlation observed with either TL or TL ratio and

proliferation rate We have added this result to the manuscript in section titled ‘**Proliferation rate is not associated to telomere length**’.

The rapid reproduction or proliferation of a cell should reduce the telomere length in dividing tumour cells. To test this, we investigated the correlation of TL with MKI67 abundance levels and a previously published proliferation score³⁰. Surprisingly, there was no association between either tumour TL ($\rho = -0.14$; $P = 0.11$) or TL ratio ($\rho = -0.09$; $P = 0.30$) and MKI67 RNA abundance (**Supplementary Fig. 3D-E**). Similarly, there was no association between proliferation scores and tumour TL ($\rho = 0.01$; $P = 0.91$) or TL ratio ($\rho = -0.05$; $P = 0.54$; **Supplementary Fig. 3F-G**). This suggests that there is a more complex relationship between proliferation and TL at play.

Supplementary Figs. 3D-E. Correlation between MKI67 RNA abundance and **D**) tumour TL and **E**) TL ratio

Supplementary Figs. 3F-G. Correlation between proliferation score and **D)** tumour TL and **E)** TL ratio.

4. The reported significant correlations are very low. Also, it is actually Spearman's ρ not Spearman ρ

We thank the reviewer for this comment and have fixed the spelling throughout the manuscript. We agree, some of the effect-sizes observed are relatively small, and have attempted to explicitly note that in the text throughout whenever appropriate.

5. In some cases the p-value is reported, it is unclear if it was adjusted for multiple testing (only reported for CNAs, Methylation, and TERT abundance). In the supplement tables, some columns mention q-values others p-value. P is always italicized.

We thank the review for the comment and have clarified which statistical assessments have been adjusted for multiple hypothesis testing throughout. In general, analyses of TL with summary statistics like PGA (**Figure 2**) or clinical covariates (**Figure 6**) are unadjusted *P* values given the very small number of tests (2-10), while genome-wide associations with TL (*e.g.* specific CNAs, or the methylation status of genes) have been FDR adjusted. *P* has been italicized throughout the manuscript.

6. How was the TL length estimated: as sum or mean of reads over all chromosomes/solely allosomes ? Is there any chance to report the TL on chromosome level ?

We thank the reviewer for this comment and have added the following description about how telomere length was estimated in the ‘**Computational telomere length estimation**’ section of the methods.

Telomere length is estimated using TelSeq and TelomerHunter. TelSeq estimates telomere length with the following formula $l = t_k s c$, where t_k is the abundance of telomeric reads (reads that contain k or more TTAGGG repeats; $k = 7$ in our study), c is a constant for the genome length divided by the number of chromosome ends and s is the fraction of all reads with GC composition between 48-52%. This represents the library size in an unbiased way, *i.e.* it takes into account sequencing technology biases. TelomerHunter also identifies telomeric reads based on the number to repeat sequences within a read and normalizes by the number of reads with GC content between 48-52%. This value is multiplied by 10^6 to calculate TRPM (telomeric reads per GC content-matched million reads) values.

As for reporting TL on a chromosome level, TelomerHunter also uses alignment information to subclassifies each read into four categories; intratelomeric, junction spanning, subtelomeric and intrachromosomal. This means that TL length can be calculated on a per chromosome level but since our sequencing is short-read sequencing we do not have many junction spanning reads and cannot accurately measure per chromosome TL.

7. Figure 2 I-J

Should be called gene fusions not just fusions.

Good point, thank you for catching this! Updated in **Figs. 2I-J**.

8. Figure 5

The Mann-Whitney test was run deleted vs non-deleted and amplified vs non-amplified. However, the red or blue dots among the black dots are very difficult to see. Maybe use of an alpha parameter can be considered.

We thank the reviewer for this comment and would like to clarify that only one Mann-Whitney test was performed per gene. The following description has been added in the methods section ‘**Association of telomere length with copy number aberrations**’ and the colours in **Figure 5** were updated to only show the class being compared

The copy number aberration state (either amplified or deleted) was determined as the status with the largest proportion of samples. Samples with aberrations in the other state were merged into the copy number neutral group. For example, if three samples had an amplification in *CHDI*, while 49 samples had a deletion. The three samples with an amplification were grouped with copy number neutral

samples and the Mann-Whitney test performed comparing the samples with and without a deletion.

Data

9. The authors analyse data from 392 patients collected from several publications, e.g. from Fraser et al., which reports 200 tumours. I cannot find a list presenting the data source and collected samples/data.

We thank the reviewer for this comment and have added the accession numbers for each of the different studies to **Supplementary Table 1** and they are now referenced in the ‘**Patient cohort**’ section of the methods

Published whole-genome sequences of tumour and matched non-tumour samples were downloaded from public repositories (phs000447.v1.p1¹¹, phs000330.v1.p1¹², EGAS00001000900¹⁴, phs000178.v11.p8¹³, EGAS00001000400²⁴, phs001648.v2.p1⁴⁴).

Method section

Computational telomere length estimation

10. In the Methods part, only TelSeq for telomere length estimation is mentioned but in the Result section also TelomereHunter was mentioned. The process description is missing in the Method, although there is a tiny description in the legend text of Figure S1. How were the results of both compared and differences considered.

We thank the reviewer for this comment and have updated the methods section ‘**Computational telomere length estimation**’ to include TelomereHunter.

Tumour and non-tumour telomere lengths were estimated using TelSeq (v0.0.1; ²⁵ and TelomereHunter (v1.0.4)²⁶ on BAM files generated using bwa-mem (version > 0.712; ⁴⁷ and GATK (version > 3.4.0; ⁴⁸).

The TL estimates from TelSeq and TelomereHunter are compared in **Supplementary Fig. 1C**. and we further investigated the correlation of TelomereHunter tumour TL estimates with recurrent prostate cancer mutations. The same clinical and prostate specific genomic associations are significant as when using the TelSeq estimates (**Reviewer Fig. 1**).

Reviewer Figure 1. Tumour TL estimates from TelomereHunter are ranked in descending order of length (top bar plot). The association of tumour TL and measures of mutational burden, TMRSS2:ERG (T2E) fusion status, as well as known prostate cancer genes with recurrent CNAs, coding SNVs, and GRs are shown. Bar plots to the right indicate the statistical significance of each association

Association of TL length with transcriptome abundance - Pathway annotation

11. They used g:Profiler but did not mention which databases they considered (KEGG, Reactome etc).

We thank the reviewer for this comment and have added the following sentence to the ‘**Over-representation analysis pathway analysis**’ section of the methods;

Pathway analysis was performed with the gprofiler2³⁵ R package using genes in which there was a significant association between TL and methylation or RNA separately using the KEGG collection of pathways³⁶

12. Just a tiny additional comment, over-representation analysis (ORA) is a very insensitive method to find enriched pathways. I would always suggest a crosstalk-based method. In the result section, the authors mention no functional enrichment for methylated genes.

We appreciate the comment by the reviewer have applied a PCA correction method to account for crosstalk as proposed in previous publications (Zhou et al., 2018; Donato et al., 2013). Although there appears to be little crosstalk between the significant pathways in our analysis, adjusting for crosstalk has a very large effect on the significance of pathways (**Supplementary Figs. 6B-C**).

Supplementary Figure 6B-C. Heatmap of crosstalk matrices where white indicates loss of significance after removal of intersecting genes

Essentially the crosstalk adjustment method, restricts and reduces the number of genes considered to be part of a pathway, so that a gene can only be represented once among all pathways. The average KEGG pathway before adjustment consisted of 64 genes, while after adjustment it only consisted of 10 (Reviewer Fig. 2).

Reviewer Figure 2. Comparison of the number of genes in a pathway before and after crosstalk adjustment.

This reduction may no longer represent the true pathway and may underestimate the enrichment. After crosstalk correction, only one out of the 32 pathways identified by gprofiler2 is still significant (**Supplementary Figs. 6D-E**).

Supplementary Figure 6D-E. Comparison of P values before and after crosstalk adjustment

Due to this, we have decided to include both the original ORA and the crosstalk adjusted analysis in the manuscript to allow readers to draw their own conclusions from a relatively conservative and a relatively liberal method as follows.

We used gprofiler2³⁴ to identify pathways enriched in genes with methylation or transcriptomic profiles that are correlated with tumour TL using KEGG pathways³⁵. We identify 16 pathways enriched in genes with methylation profiles and 16 pathways that were enriched in genes with transcriptomic profiles that were correlated with tumour TL (**Supplementary Fig. 6A**). To reduce false positives and account for crosstalk between pathways, we applied a crosstalk correction method^{36,37}. The crosstalk matrix (**Supplementary Fig. 6B**) identified overlap between the cancer related pathways, and after crosstalk adjustment only one pathway remained enriched in genes with transcriptomic profiles that were correlated to tumour TL: hsa04519 Focal adhesion (**Supplementary Fig. 6C**).

Reviewer #2

This is a nice multi-omic study that brings several strands together ie genomic complexity, telomerase expression, methylation patterns and clinical features to provide further evidence for a role in telomere dysfunction and genomic complexity in prostate cancer. However it is a descriptive study and it wasn't clear what key new information is provided, as such it appears to be a somewhat incremental.

I have no major issues with the data as it is presented.

Minor points:

1. Supplementary figure 9, indicates a U-shaped distribution for HRs as a function of telomere length in the tumour samples. What is the significance of this? Is this consistent with previous reports (Svenson et al Tumour Biol. 2017; Renner et al. Prostate Cancer 2018) that long telomere length in blood is associated with poor survival.

We thank the reviewer for the comment and agree that this observation is interesting. This suggests that both long and short tumour TLs are related to biochemical relapse, although not significantly in our models. Long tumour TL may protect the tumour from replication senescence and allow more driver mutations to accumulate. Shorter tumour TLs may expose chromosome ends, facilitating breakage-fusion-bridge cycles, leading to high genomic instability. This is significant because any dysregulation of telomere lengths is detrimental to the cell. Svenson *et al.* found that relative TL was associated with prostate cancer specific and metastasis-free survival and Renner *et al.* looked at overall mortality which are different end points than our study which looked at biochemical relapse. Our data suggests that shorter normal TL and smaller (less negative) \log_2 (TL ratio) are associated with a higher rate of BCR. This might reflect the differences in the cohort (NCCN intermediate vs. high risk) or the different end points.

To facilitate the interpretation of our biochemical relapse analysis, we have made the comparisons in our cut-point analysis consistent, so that the group with the longer TLs is always the reference group. The HRs displayed in **Figure 6I-J** are now more consistent with the HRs in **Supplementary Figure 9A-C**.

Figure 6 | I-K, Cox proportional hazard models were created for **I**, non-tumour (blood) TL, **J**, tumour TL and **K**, TL ratio with BCR as the endpoint

Supplementary Figure 9 | A-C, Association of **A**, non-tumour TL, **B**, tumour TL and **C**, $\log_2(\text{TL ratio})$ with BCR using a Cox proportional hazards model at different TL cutoffs, incremented by 50 bp.

2. State in methods why Spearman’s correlation was chosen over Pearson’s

We thank the reviewer for this question. We chose to use Spearman’s rank correlation because it is a nonparametric test and does not assume that values are sampled from a population that follows a Gaussian distribution or that there is a linear relationship between the variables. As well, comparing the ranks handles outliers better and can identify monotonic relationships. We have added this information to the methods in the ‘**Statistical analyses and data visualization**’ section.

3. Line 114-115 - statement that ‘samples with *CHD1*, *RB1* or *NKX3-1* deletions had shorter tumour TL (Fig. 1D)’ is not apparent from the data shown in Fig 1 D. They should provide a data plot that illustrates this point or remove this statement.

We thank the reviewer for this comment. Boxplots showing the difference in tumour TL and TL ratio for known prostate cancer genes (including in *CHD1*, *RB1* or *NKX3-1*) can be seen **Fig. 5**. We have changed the sentence in the manuscript to the following, to draw this fact to the reader’s attention.

While tumour TL was not associated with any known prostate cancer-related genomic rearrangement (GR) or single nucleotide variant (SNV) at current

statistical power, samples with *CHD1*, *RB1* or *NKX3-1* deletions had shorter tumour TL (Fig. 1D; Fig. 5A).

Figure 5 | A, Difference in tumour TL between samples with a copy number aberration and those without in prostate cancer related genes and associated genes. **B**, Difference in TL ratio between samples with a copy number aberration and those without in prostate cancer related and associated genes. *Q* values are from a Mann-Whitney U test and are bolded when significant ($Q < 0.05$). Colour of the dots indicate copy number status of the gene: amplification (red), deletion (blue), or neutral (black). Boxes with a white background are known prostate cancer genes, while boxes with a gray background were identified by a genome wide search.

Reviewers' Comments:

Reviewer #1:

Remarks to the Author:

Thank you for the clear replies.

I have no further comments

Reviewer #2:

Remarks to the Author:

The authors have adequately addressed my comments and the revised manuscript has improved.